# Development of a New Enzyme-Linked Immunosorbent Assay (ELISA) for Measuring the Content of PACAP in Mammalian Tissue and Plasma

**DOI:** 10.3390/ijms242015102

**Published:** 2023-10-12

**Authors:** Elisabeth Anne Adanma Obara, Birgitte Georg, Jens Hannibal

**Affiliations:** Department of Clinical Biochemistry, Faculty of Health Sciences, Bispebjerg and Frederiksberg Hospital, University of Copenhagen, Nielsine Nielsens Vej 4, 2400 Copenhagen, NV, Denmark; birgitte.georg@regionh.dk (B.G.); jens.hannibal@regionh.dk (J.H.)

**Keywords:** PACAP, ELISA, assay validation, migraine

## Abstract

Pituitary adenylate cyclase-activating polypeptide (PACAP) is a naturally occurring neuropeptide found in both the central and peripheral nervous systems of vertebrates. Recent studies have revealed the presence of PACAP and its corresponding receptors, namely, the pituitary adenylate cyclase-activating polypeptide type I receptor (PAC1R), vasoactive intestinal peptide receptor 1 (VIPR1), and vasoactive intestinal peptide receptor 2 (VIPR2), in various structures implicated in migraine pathophysiology, including sensory trigeminal neurons. Human studies have demonstrated that when infused, PACAP can cause dilation of cranial vessels and result in delayed migraine-like attacks. In light of this, we present a novel ELISA assay that has been validated for quantifying PACAP in tissue extracts and human plasma. Using two well characterized antibodies specifically targeting PACAP, we successfully developed a sandwich ELISA assay, capable of detecting and accurately quantifying PACAP without any cross-reactivity to closely related peptides. The quantification range was between 5.2 pmol/L and 400 pmol/L. The recovery in plasma ranged from 98.2% to 100%. The increasing evidence pointing to the crucial role of PACAP in migraine pathophysiology necessitates the availability of tools capable of detecting changes in the circulatory levels of PACAP and its potential application as a reliable biomarker.

## 1. Introduction

Pituitary adenylate cyclase-activating polypeptide (PACAP) belongs to the vasoactive intestinal peptide (VIP)/glucagon/secretin family of peptides and has a broad range of biological functions owing to its pleiotropic nature [1]. PACAP shares significant sequence homology (68%) with VIP. The discovery of PACAP dates as far back as 1989, where it was initially isolated from ovine hypothalamus due to its ability to activate adenylate cyclase and thereby induce cyclic adenosine monophosphate (cAMP) in pituitary cells [2]. It has since been shown to exist in two isoforms: the highly abundant PACAP 38 consisting of 38 amino acids and the C-terminally truncated form PACAP 27, consisting of 27 amino acids, which only accounts for 10% of total PACAP in vertebrates [2,3,4,5].

PACAP acts both as a neurotransmitter and neurotrophic factor via three G-protein coupled receptors: (1) pituitary adenylate cyclase-activating polypeptide type I receptor (PAC1R) considered to be PACAP specific, (2) vasoactive intestinal peptide receptor 1 (VIPR1) and (3) vasoactive intestinal peptide receptor 2 (VIPR2), which share equal affinity for both PACAP and VIP [6,7,8]. PACAP and its receptors, PAC1R, VIPR1 and VIPR2, are expressed throughout the central and peripheral nervous system [9,10,11,12,13]. In human studies, PACAP has been shown in perivascular nerves innervating pial arteries as well as in nerve fibers and neurons in several brain regions including the hypothalamic, preoptic and brainstem nuclei [14,15,16]. More extensive tracing studies in rodents have shown PACAP gene expression immunoreactivity in the cortex, hippocampus, brain stem and cerebellum, with highest expression in the hypothalamic region [5,17,18]. Although PACAP is highly expressed in the central nervous system, it is also found in the peripheral sensory and para-sympathetic nervous system in tissues such as reproductive organs, respiratory tract, gastro-intestinal tract, endocrine glands, retina, and the pancreas [19,20].

The amino acid sequence of PACAP 38 is highly conserved in vertebrates, most likely due to the central role of PACAP in regulating a variety of biological processes [19]. Functional studies have shown PACAP to play a role in anti-apoptosis, anti-inflammation, neural development, glial regulation, and retinal protection [21,22,23,24,25,26]. Evidence has accumulated implicating PACAP, like calcitonin gene-related peptide (CGRP), in migraine pathophysiology. Anatomical structures related to migraine such as the sensory trigeminal ganglions, trigeminal nucleus caudalis, and perivascular nerve fibers express PACAP and its receptors [27,28]. Like CGRP, intravenous administration of PACAP triggers vasodilation of cranial arteries [29,30]. Furthermore, changes in the circulatory levels of PACAP have been shown to induce migraine. To date, few studies have shown the kinetics of circulatory PACAP and the levels of PACAP in circulation required to induce migraine attacks [31,32].

To elucidate the biological function of PACAP, it is essential to be able to measure accurately and specifically, the physiological levels of PACAP in both tissue and plasma samples. Here, we introduce a novel highly specific and sensitive sandwich enzyme-linked immunoassay (ELISA) based on two well characterized antibodies. The multifaceted ELISA is validated for measuring PACAP in both tissue extracts from rodents and in human plasma samples.

## 2. Results and Discussion

### 2.1. Limits of Detection and Quantification

The calibration curve shown in Figure 1, was derived from six serial dilutions of synthetic PACAP 38 ranging from 5.12 pmol/L to 400 pmol/L. From ten independent plates, the accuracy of the calibration curve was within the accepted range and the %CV was <20% as shown in Table 1. The lower limit of quantification (LLOQ) and upper limit of quantification (ULOQ) were defined respectively as the lowest and highest concentrations of PACAP 38 that can be accurately quantified to give a recovery of 100 ± 25% and %CVs below 20%. The two points defining the range of quantification 400 pmol/L and 5.12 pmol/L were assayed 10 times in three separate runs. The %accuracy and %CV were both within an acceptable range as shown in Table 2. All samples beyond the limits of the LLOQ and ULOQ were defined as not determined (n/d) for downstream runs. The limit of detection (LOD) is the lowest concentration in which the designed ELISA can distinguish PACAP 38 from a blank sample. The LOD for the sandwich ELISA was 3.4 pmol/L (Appendix A). 

### 2.2. Precision

The precision of the PACAP 38 ELISA sandwich assay was assessed by determining the levels of PACAP 38 in the quality control (QC) samples and mouse tissue extracts. 

Intra- and inter-run precision of quality control (QC) samples.

The mean, SD and %CV for each QC sample was determined from three identical plates with all readouts falling between the LLOQ (5.12 pmol/L) and ULOQ (400 pmol/L). As summarized in Table 3, the %CV for QC1 and QC2 were <20%, indicating acceptable intra-run precision of the assay. The inter-run precision was determined from nine plates with all samples assessed in duplicate on each plate. The two QC samples had a %CV < 20% and an accuracy of 111% for QC1 and 116% for QC2.

#### Intra- and Inter-Run Precision of Mouse Tissue Extracts

The levels of PACAP 38 in the four brain regions namely the cortex, cerebellum, brainstem, and hypothalamus of PACAP wildtype (WT) and PACAP deficient (KO), were analyzed to ascertain both intra and inter assay precision. 

For intra-run precision, the levels of PACAP 38 in the indicated regions of the brain were run by two researchers on the same day each using three identical plates for each mouse pair (PACAP 38 WT and KO). As shown in Figure 2 and Table 4, high levels of PACAP 38 were observed in the hypothalamus and brainstem compared to the cortex and cerebellum of WT animals while PACAP could not be detected by the ELISA in extracts of KO animals (Appendix A). Immunohistochemistry staining for PACAP 38 confirmed the differences in expression showed in the ELISA. As shown in Figure 3b, the hypothalamus expressed relatively high levels of PACAP compared to the other brain regions (cortex, cerebellum, and brainstem) (Figure 3a,c,d). The PACAP^−/−^ mice had undetectable levels of PACAP 38 (See Appendix A). The intra-run precision for the cortex, cerebellum, brainstem, and hypothalamus for WT1, WT2 and WT3 based on the %CV were <20% indicating acceptable repeatability and reproducibility of the ELISA within a daily run (Table 4). The inter-run precision was assessed using tissue extracts from WT4. Samples from the cortex, cerebellum, brainstem, and hypothalamus were run on 9 plates, on non-consecutive days by two researchers. As shown in Table 5, the %CV for all samples were <20% indicating a good repeatability and reproducibility across multiple days.

### 2.3. Recovery

Recovery of PACAP 38 in human plasma samples was assessed by the ability of the novel sandwich ELISA to detect known amounts of PACAP 38. Samples with undetectable amounts of PACAP obtained from healthy controls were spiked with synthetic PACAP 38 to yield 400 pmol/L and assessed on three plates in quadruplet. As shown in Table 6, the overall recovery for of PACAP 38 in PL1 and PL2 were 97.4% and 101% respectively. The recovery and %CV were within an acceptable range of 100 ± 25% and <20% respectively. 

### 2.4. Dilution Linearity

Dilution linearity was assessed to determine whether the novel ELISA could reliably quantify the levels of PACAP after serial two-fold dilutions throughout the standard curve using both spiked human plasma samples and tissue extracts from the hypothalamus of PACAP WT mice. To determine linearity, the spiked plasma samples were serially diluted and run on three plates on non-consecutive days. The recovery of synthetic PACAP 38 in PL1 and PL2 of 106% and 103% for the 1:2 dilution, 116% and 107% for the 1:4 and 117% and 102% for the 1:8 dilution (Figure 4a). Dilution linearity was also assessed in mouse tissue extracts of the hypothalamus to assess the binding capacity to endogenous PACAP after serial two-fold dilutions. The accuracy was 98% and 94% in the 1:10 dilution and 93% and 87% in the 1:20 dilution for WT1 and WT2 respectively (Figure 4b).

### 2.5. Specificity

Specificity was assessed by the ability to detect the truncated isoform, PACAP 27, and peptides in the VIP/glucagon/secretin family. Measurements of PACAP 27 was done at three known concentrations (100 pmol/L, 50 pmol/L and 25 pmol/L) (Table 7) on fifteen different plates, showing the ELISA’s specificity to detect and quantify PACAP 27. The accuracy (%) and %CV were within the accepted range of 100 ± 25% and <20% respectively. On the contrary, readouts below the LLOQ were observed for other peptides belonging to the VIP/glucagon/secretin family assayed at a concentration of 200 pmol/L (Table 8). 

Since the discovery of PACAP in extracts from ovine hypothalamus in 1989 [2], studies have established the feasibility of quantifying the levels of PACAP by ELISA and radioimmunoassay (RIA) based assays in several types of samples including tissue extracts from rodents and human plasma samples. Most of these assays are dependent on based assays in several types of samples including tissue extracts from rodents and human plasma samples. Most of these assays are dependent on specific PACAP antibodies, validated for use in either RIA or enzyme immunoassays (EIA/ELISA) [5,17,33,34,35]. However, the success of utilizing these methods is limited due to the availability of specific antibodies and for RIA assays difficulty in handling radioactivity. The assessment of PACAP in plasma has also been quantified using Liquid chromatography coupled with tandem mass spectrometry (LC-MS/MS) [36].

In this study, we opted to develop and validate a quantitative sandwich ELISA assay capable of precisely measuring low concentrations of PACAP in diverse matrices. The novel PACAP ELISA assay designed in this study showed a sensitivity comparable to our previously described RIA in analysis of tissue extracts [5,37]. We compared the content of PACAP in different regions of the mouse brain (hypothalamus, cortex, cerebellum, and brain stem). We found similar to the rat brain, the hypothalamus contained the highest amount of immunoreactive PACAP compared to the other regions, the brain stem being the second highest of the measured regions. This corresponds well with the observed localization of PACAP immunoreactivity as found in rat brain and mouse brain (Figure 3a–d) [18]. Another study by Masuo et al. showed comparable distribution pattern of PACAP in the rat brain with the highest concentration detected in the hypothalamic regions [16]. 

The precision analysis of the assay was used to determine the repeatability and reproducibility. Intra run precision ranged between 2.2% and 14.0% for both quality control samples and mouse tissue extracts. Inter run precision ranged between 9.4 and 17.5. These results are within an acceptable range of <20% for both repeatability and reproducibility, indicating the assay is reliable and independent of the analyst or day to day variations.

The new assay utilizes a well characterized antibody pair consisting of a mouse monoclonal anti-PACAP antibody with high affinity to PACAP 6-18 recognizing both PACAP 27 and PACAP 38 and a polyclonal rabbit anti-PACAP antibody with high affinity for both PACAP 27 and PACAP 38 [5]. As demonstrated, the assay has high sensitivity comparable to a previously established in house RIA showing high specificity to PACAP with no cross reactivity with other structurally related peptides belonging to the VIP/secretin/glucagon family [5]. 

Measurements of PACAP in human plasma/serum have demonstrated different concentrations of “PACAP-like” immunoreactivity. Infusion of PACAP leads to rapid decrease in blood pressure due to the relaxing effect on smooth muscle, tachycardia and flushing of the skin [37]. However, clearance of endogenous PACAP or after infusion in circulation during normal conditions is not well known. Dipeptidyl peptidase IV (DP IV/CD26), an enzyme involved in neuropeptide inactivation was proposed to play a critical role in the degradation of PACAP in circulation. [36,38]. A study by Zhu et al. showed a significant decrease in the clearance of PACAP and its degradation product, the inactive PACAP-(3-38), in mice lacking DP-IV, [36]. In circulation, PACAP has also been shown to be bound to ceruloplasmin, which possibly renders it inactive, hence requiring extraction of PACAP from ceruloplasmin prior to analysis [39]. In the current PACAP ELISA, pre-treatment by extraction of plasma was not required to detect the total amount of PACAP in plasma, a procedure that may lead to loss of PACAP and thereby induce variability [36,37]. 

In the validation experiments, spiked human plasma samples were used to test the recovery PACAP. Although spiked samples do not reflect the complexity of clinical samples, with reference to expected levels of PACAP, and storage stability, we present an assay that covers the expected range of PACAP in migraineurs and after infusion of PACAP in both healthy controls and migraineurs [32,37]. 

## 3. Materials and Methods

### 3.1. Reagents and Buffers

The following synthetic peptides were used for the development and validation of the PACAP 38 sandwich ELISA: PACAP 38 (Cat. no. H-1132) and PACAP 27 (Cat. no. H-1172) were purchased from Bachem AG. PACAP—related peptide (PRP) (Cat. no. 8974), peptide histidine isoleucine-27 (PHI-27) (Cat. no. 7146), peptide histidine-methionine (PHM-27) (Cat. no. 7176), Secretin (Cat. no. 7314) and Glucagon (Cat. no. 7165) were purchased from Peninsula Laboratories (Belmont, CA, USA). 

### 3.2. Antibodies

This ELISA is based on two previous well characterized in house antibodies. The monoclonal anti-PACAP antibody (mAb JHH1), which recognizes an epitope between the amino acid 6–16 of PACAP thus binding both PACAP 27 and PACAP 38, was used as a capture antibody [5]. The rabbit anti-PACAP antibody (pAb 523C) previously characterized to bind epitopes located between amino acid 16–38 of PACAP, was used as a detection antibody [5].

### 3.3. Immunohistochemistry

Immunohistochemistry was performed on mouse brain tissue sections as described previously [40]. Briefly, sections were treated by antigen retrieval at 80 °C for 1.5 h, then 1% hydrogen peroxide (H_2_O_2_) to block endogenous peroxidase activity, thereafter sections were blocked in 5% donkey normal serum and incubated with the primary antibody, pAb 523C diluted at 1:150,000, overnight at 4 °C. The following day, sections were incubated sequentially with biotinylated tyramide and an avidin–biotin–peroxidase reagent (ABC Elite; Vector Labs, Newark, CA, USA) to amplify the PACAP 38 signal then visualized with Alexa Fluor A488-conjugated donkey IgG. 

### 3.4. Sandwich ELISA Protocol

The protocol used for the development of the PACAP sandwich ELISA is as follows. In brief, clear, flat bottom 96 well plates (Cat no. CLS9018; Corning Merck, Merck KGaA, Darmstadt, Germany) were precoated with the capture monoclonal antibody PACAP (mAb JHH1) at 0.5 mg/mL in 0.2 M bicarbonate (NaCO_3_ and NaHCO_3_) buffer (pH 9.5) and incubated overnight at 4 °C. The coating buffer was discarded, the plates were washed 3 times using PBST (4 mM phosphate-buffered saline with Tween^®^ 20 (Cat.no: P1379; Sigma, Merck KGaA, Darmstadt, Germany)), then all free binding sites were blocked in PBST supplemented with 5% skimmed milk for 3 h at room temperature. Following incubation with blocking buffer, the plates were washed, then the samples and standards prepared in assay buffer were added to the plates in duplicates and incubated overnight at 4 °C. Then, plates underwent sequential addition of the detection antibody (pAb 523C) and the secondary biotinylated antibody (Cat no. 711-065-152; Jackson Immuno Research), both incubated at 4 °C overnight. For detection, the signal was amplified by incubation with an Avidin Biotin complex (ABC) (Vector Laboratories) according to the kit instructions before incubation with 3,3′,5,5′-Tetramethylbenzidine (TMB) (Sigma Aldrich, St. Louis, MI, USA). The enzymatic reaction with TMB was quenched by addition of 2 M H_2_SO_4_ (Sigma Aldrich) and the absorption was read at 450 nm using a Tecan microplate reader (M200 Tecan, Nordic, Mölndal, Sweden). A standard curve was extrapolated using a 4-parameter logistic (4-PL) curve fit for determining the concentration of PACAP 38 in the samples.

### 3.5. Animals

All animals used in this study were bred inhouse at the Department of Clinical Biochemistry at Bispebjerg and Frederiksberg hospital, Denmark. Mice were housed in isolated ventilated cages under 12:12 light/dark cycles with food (Altromin 1324; Altromin Spezialfutter, Lage, Germany) and water ad libitum. All mice procedures were performed in accordance with the principles of Laboratory Animal Care (Law on Animal Experiments in Denmark) and under Danish Veterinary Authorities (Dyreforsoegstilsynet) license. A formal waiver was approved to Jens Hannibal under license number 2021-15-0201-00929 designated by the Danish Animal ethics committee. A total of 8 mice of both sexes, 4 PACAP wildtype (WT 1–4) and 4 PACAP deficient (KO 1–4), age 8–15 weeks were included in the study. 

#### Mouse Sample Preparation

For tissue extraction, mice were euthanized by decapitation. The following areas of the central nervous system were collected: brainstem, cerebellum, cortex, and hypothalamus (Figure 3a–d). Initially, the brainstem was collected, then the cerebellum was quickly cut at a coronal plane, both tissues were frozen on dry ice before collection of the cortex and hypothalamus. The two brain hemispheres were mounted on a vibratome filled with ice cold Hanks’ buffer. Three consecutive 500 µm coronal sections were carefully cut at the level of the suprachiasmatic nucleus (SCN) to allow for orientation and to ensure repeatability of the assay. The hypothalamus and cortex on all three sections were carefully dissected, then pooled before snap freezing on dry ice. Prior to assessing the levels of PACAP by ELISA all tissues were weighed and sequentially homogenized in an appropriate volume of boiling distilled water, and ice-cold acetic acid as described previously [5]. The homogenate was centrifuged, and the supernatant dried overnight using a HetoVac. Before analysis, the dried sample was reconstituted in an appropriate volume of assay buffer and analyzed by ELISA.

### 3.6. Human Plasma Sample Preparation

Blood samples from healthy donors (PL1 and PL2) were collected in tubes containing 50 IU of heparin and 500 kIU of aprotinin per ml of blood [37]. The samples were then centrifuged for plasma collection at 1500× *g* at 4 °C before storage at −20 °C for subsequent analysis by ELISA. 

### 3.7. Assay Validation

#### 3.7.1. Calibration Standards and Quality Controls

The calibration curve and quality control samples were prepared using synthetic PACAP 38 in assay buffer. To generate a six-point standard curve, serial dilution of synthetic PACAP38 was used at the following concentrations: C1 = 400 pmol/L, C2 = 200 pmol/L, C3 = 80 pmol/L, C4 = 32 pmol/L, C5 = 12.8 pmol/L, C6 = 5.12 pmol/L. Two quality control samples, QC1 = 100 pmol/L and QC2 = 15 pmol/L were used as positive controls. The quality controls were aliquoted and stored as single ready to use vials at −80 °C.

#### 3.7.2. Limits of Detection and Quantification

The lower limit of quantification (LLOQ) and upper limit of quantification (ULOQ) were defined as the extremes of the standard curve in which the accuracy was within an acceptable range of 100 ± 25%. Each point on the calibration curve was assessed comparing the nominal concentration to the observed concentration (accuracy %).
Accuracy % = (observed value)/(expected value) × 100

The limit of detection (LOD) was defined as the mean O.D. of blank plus 3 standard deviations (SD). A CV < 20% was accepted.

#### 3.7.3. Precision Analysis

Intra-assay precision was assessed using both quality control samples (QC1 and QC2), and mouse tissue extracts from WT1, WT2 and WT3, by two independent researchers on the same day on three identical plates. The samples used represented high, medium, and low levels of PACAP 38, enabling good coverage of the standard curve. Inter-assay precision was assessed using both quality control samples (QC1 and QC2) and mouse tissue extracts from WT4 on nine independent plates, run on non-consecutive days by two researchers. A CV < 20% was accepted. 

#### 3.7.4. Recovery of PACAP in Human and Mouse Plasma Samples

Recovery experiments were run in human plasma samples form healthy donors. Human plasma samples were spiked with PACAP 38 giving a final concentration of 400 pmol/L. For both samples %-recovery of 100 ± 25% was accepted and calculated as follows:Recovery % = (observed value)/(expected value) × 100

#### 3.7.5. Dilution Linearity

Linearity was assessed using both spiked human plasma and mouse tissue extracts of the hypothalamus. Spiked plasma samples were serially diluted two-fold and analyzed for accuracy and CV across three plates. Mouse tissue extracts obtained from the hypothalamus, which show relatively high levels of PACAP 38 [5] were serially diluted two-fold and analyzed for accuracy and CV relative to the original sample. All sample dilutions were run in four wells per plate across three plates. A CV < 20% was accepted.

#### 3.7.6. Specificity

Specificity was assessed by testing for cross reactivity with peptides in the VIP/glucagon/secretin family namely VIP, Glucagon, PHM-27, Secretin, PRP and PHI-27 at a high concentration of 200 pmol/L. In mice samples, specificity was assessed using tissue extracts from mice lacking PACAP (KO 1–4). All experiments were performed by two separate researchers on non-consecutive days.

### 3.8. Statistical Analysis

All statistical analysis were carried out using GraphPad Prism 9.0 (GraphPad Software Inc., La Jolla, CA, USA) Data analysis was carried out using Excel 2016 (Microsoft Corporation, Redmond, WA, USA).

## 4. Conclusions

In conclusion, the present study describes a newly developed sandwich PACAP ELISA validated for quantification of PACAP in tissues extracts and human plasma. The rigorous validation of this new ELISA detecting PACAP in samples from both primates and rodents shows that it meets all requirements for robust and reliable measurements. We could show comparable levels of PACAP in the mouse brain as previously published assays, and high recovery in measurement of PACAP in spiked human plasma samples. However, further studies would be of interest to investigate the capacity of the proposed assay in assessing PACAP kinetics in plasma and the role of PACAP for instance in migraine.

## Figures and Tables

**Figure 1 ijms-24-15102-f001:**
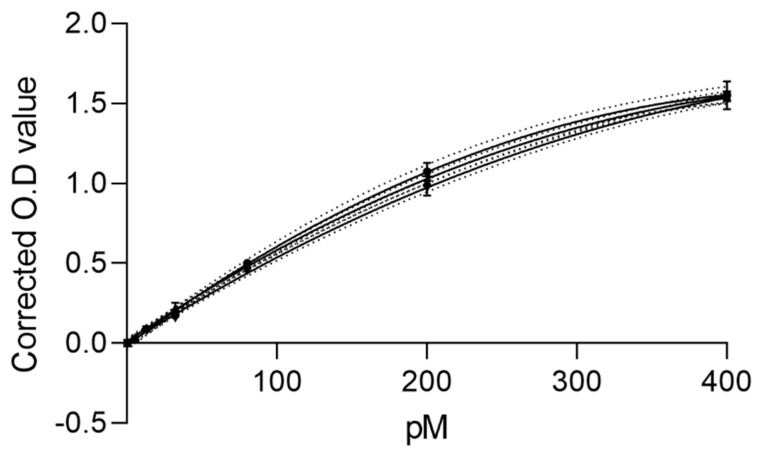
Representative standard curves for PACAP 38 ELISA assay (*n* = 3). The graph was generated by combining serial dilutions of synthetic PACAP 38 in range 5.12–400 pmol/L and interpolated with GraphPad Prism v. 9.0 software by a four-parameter regression formula. The error bars show the standard deviation between the duplicate samples in three independent runs. *y* axis: Optical Density (O.D), *x* axis: PACAP 38 concentration (pmol/L).

**Figure 2 ijms-24-15102-f002:**
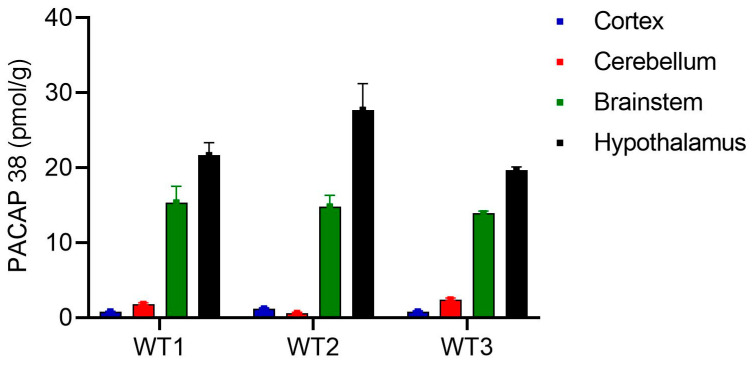
Representative distribution of PACAP 38 in the cortex, cerebellum, brainstem and hypothalamus of PACAP wildtype (WT 1, WT2 and WT3) mice.

**Figure 3 ijms-24-15102-f003:**
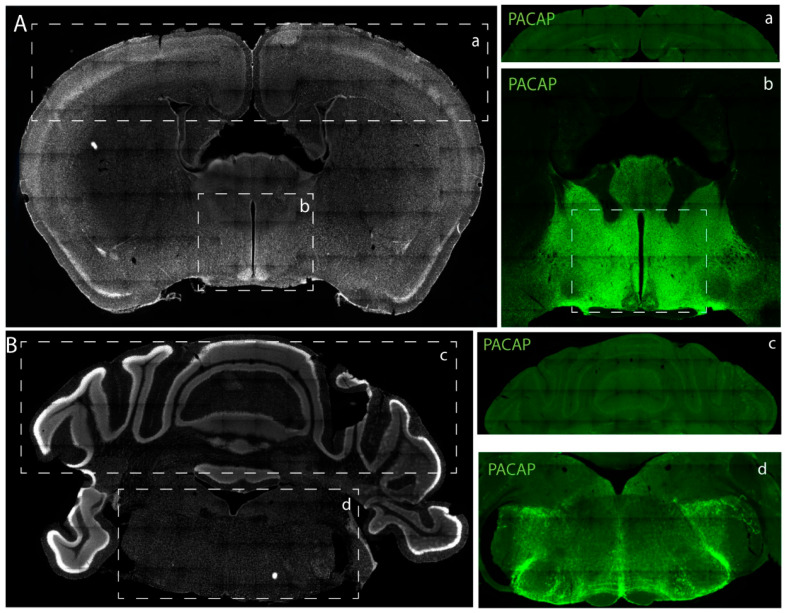
PACAP expression in the mouse brain: (**A**) Tissue section showing the (**a**) cortex and (**b**) hypothalamus. (**B**) Tissue section showing the (**c**) cerebellum and (**d**) brainstem. Representative immunohistochemical staining for PACAP (green) in the (**a**) cortex, (**b**) hypothalamus, (**c**) cerebellum and (**d**) brainstem.

**Figure 4 ijms-24-15102-f004:**
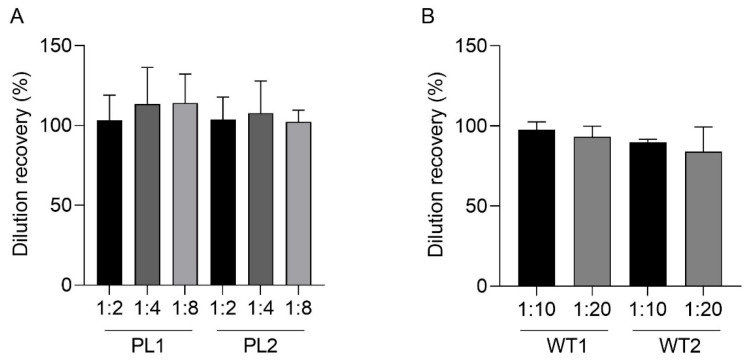
Bar graph showing dilution linearity of spiked human plasma samples and mouse tissue extract. Dilution recovery ((%), *y*-axis) of (**A**) two spiked human plasma samples (PL1 and PL2), followed by results of serial dilution at 1:2, 1:4 and 1:8 in assay buffer and (**B**) hypothalamus tissue extracts from wildtype mice (WT1 and WT2), followed by serial dilution at 1:10 and 1:20. The error bars indicate the mean and standard deviation (SD) of each sample after three independent runs. Dilution recovery was deemed acceptable at 100 ± 25%.

**Table 1 ijms-24-15102-t001:** Statistical data of back-calculated concentrations of PACAP, from 10 independent runs using six calibration points.

Expected Concentration (pmol/L)	Observed Concentration (pmol/L)			Accuracy (Observed/ Expected) × 100 (%)
	Mean	SD	%CV	
400	404	4.3	1.1	101
200	195	4.0	2.0	97.6
80	85.2	4.0	4.7	107
32	32.2	2.6	8.0	101
12.8	13.5	1.1	8.2	105
5.12	3.84	0.6	15.7	75.1

CV: Coefficient of variation; SD: standard deviation, *n* = 10 plates.

**Table 2 ijms-24-15102-t002:** PACAP ELISA assay specification showing the ULOQ and LLOQ.

Expected Concentration (pmol/L)	Observed Concentration (pmol/L)			Accuracy (Observed/ Expected) × 100 (%)
	Mean	SD	%CV	
400	375	29.2	7.8	93.8
5.12	4.5	0.6	13.0	87.4

CV: Coefficient of variation; SD: standard deviation, *n* = 3 plates.

**Table 3 ijms-24-15102-t003:** Intra- and inter-assay precision of PACAP38 ELISA by determining the levels of PACAP in quality control samples.

Precision Assay	Statistics	PACAP 38 Concentration (pmol/L)	
Intra-assay ^a^		QC1 (100)	QC2 (15)
	Mean	101	15.9
	SD	8.6	2.1
	%CV	8.6	13.1
	Accuracy (observed/expected) × 100 (%)	98	98
Inter-assay ^b^	Mean	111	17.4
	SD	12.9	3.1
	%CV	11.7	17.5
	Accuracy (observed/expected) × 100 (%)	111	116

CV: Coefficient of variation; SD: standard deviation, ^a^
*n* = 3 plates; ^b^
*n* = 9 plates.

**Table 4 ijms-24-15102-t004:** Intra-assay precision of PACAP 38 ELISA by determining the levels of PACAP 38 in mouse brain tissue extracts.

ID: WT1 ^a^	PACAP 38 Concentration (pmol/g)			
	Cortex	Cerebellum	Brainstem	Hypothalamus
Mean	0.8	1.8	15.4	21.7
SD	0.04	0.2	2.2	1.7
%CV	5.5	8.4	14.0	7.8
**ID: WT2 ^a^**	**PACAP 38 concentration (pmol/g)**			
	Cortex	Cerebellum	Brainstem	Hypothalamus
Mean	1.2	0.6	14.8	27.7
SD	0.1	0.1	1.5	3.5
%CV	6.8	13.1	9.9	12.5
**ID: WT3 ^a^**	**PACAP 38 concentration (pmol/g)**			
	Cortex	Cerebellum	Brainstem	Hypothalamus
Mean	0.8	2.4	13.9	19.7
SD	0.1	0.2	0.3	0.4
	6.7	7.8	2.2	2.2

CV: Coefficient of variation; SD: standard deviation, PACAP wildtype mouse (WT1–3), ^a^
*n* = 3 plates.

**Table 5 ijms-24-15102-t005:** Inter-assay precision of PACAP 38 ELISA by determining the levels of PACAP 38 in mouse brain tissue extracts.

ID: WT4 ^a^	PACAP 38 Concentration (pmol/g)			
	Cortex	Cerebellum	Brainstem	Hypothalamus
Mean	0.8	4.3	14.5	26.7
SD	0.1	0.4	1.8	2.6
%CV	15.1	9.4	12.4	9.7

CV: Coefficient of variation; SD: standard deviation, PACAP wildtype mouse (WT4), ^a^
*n* = 9 plates.

**Table 6 ijms-24-15102-t006:** Recovery of synthetic PACAP 38 in human plasma samples.

	PACAP 38 Concentration (pmol/L)	
	PL1	PL2
Mean	390	403
SD	57.9	42.0
%CV	14.9	10.4
Recovery (%)	97.4	101

CV: Coefficient of variation; SD: standard deviation, human plasma (PL1 and PL2), *n* = 3 plates.

**Table 7 ijms-24-15102-t007:** Specificity of the PACAP 38 ELISA in determining PACAP 27.

PACAP 27 Concentration (pmol/L)	Observed Concentration (pmol/L)			Accuracy (Observed/ Expected) × 100 (%)
	Mean	SD	%CV	
100	120	12.5	10.4	120
50	48.1	5.7	11.9	96.2
25	19.3	2.0	10.5	77.2

CV: Coefficient of variation; SD: standard deviation, *n* = 15 plates.

**Table 8 ijms-24-15102-t008:** Specificity of the PACAP 38 ELISA in determining cross reactivity with peptides in the VIP/glucagon/secretin family.

Analyte	Specificity (%)
PACAP 38	100
PACAP 27	100
VIP ^a^	n/d
PRP ^a^	n/d
PHI-27 ^a^	n/d
PHM-27 ^a^	n/d
Secretin ^a^	n/d
Glucagon ^a^	n/d

CV: Coefficient of variation; SD: standard deviation, ^a^
*n* = 5 plates, n/d: not determined.

## Data Availability

The data presented in this study are available on request from the corresponding author.

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
