# Peer review of "Development of a New Enzyme-Linked Immunosorbent Assay (ELISA) for Measuring the Content of PACAP in Mammalian Tissue and Plasma"

_ijms, 2023, doi:10.3390/ijms242015102_

Round 1

Reviewer 1 Report

Authors described the assay development and qualification of an ELISA for teh detection of PACAP 38, one of the neurotransimitters in CNS. The paper is well written with sufficient introduction and experimental details. I suggest the paper to be accepted by doing the following revisions

1. In the recovery section, since PACAP is abundant in CNS, authors should use the brain tissue extract to evaluate the recovery so as to avoid the sample matrix effect. 

2. Figure S1 should be labeled as main figure in main text since it appears in main text. Otherwise, it should be removed from the manuscript and put it in the SI.

Author Response

We are very grateful for the opportunity to revise our manuscript. Hereby, we would like to confirm that that all relevant changes have been highlighted in yellow in the new version of the manuscript to visualise them. We would like to thank you for proposing our article for submission.

COmment 1: 

: We agree with the reviewer that it would be interesting to test the recovery of PACAP in mouse tissue from PACAP deficient mice. We ran a test to highlight that we see the same recovery in mouse plasma samples as we see using human plasma. However, we have opted not to include this data in the article. Below we have an overview of the results using mouse plasma for validation of the recovery of the ELISA assay. The recovery is coherent with that seen in human plasma samples.

Recovery of synthetic PACAP 38 in mouse plasma samples.

PACAP 38 concentration (pmol/L)

mPL1

mPL2

Mean

112

127

SD

5.7

1.2

%CV

5.1

0.9

Recovery (%)

112

127

CV: Coefficient of variation; SD: standard deviation, an=3 readouts

Response to Comment 2: 

We agree with the reviewer and have moved Supplementary Figure 1 to the Main text as Figure 3

Reviewer 2 Report

 The manuscript entitled Development of a new Enzyme-Linked Immunosorbent Assay 2(ELISA) for measuring the content of PACAP in mammalian tissue and plasma .The article is well written, however, there are some major and minor comments that need to be addressed. The main comment concerns In light of this, it presents a novel ELISA assay that has been validated for quantifying PACAP in tissue extracts and human plasma. Using two well characterized antibodies specifically targeting PACAP,  developed a sandwich ELISA assay, capable of detecting  and accurately quantifying PACAP without any cross-reactivity to closely related peptides. The quantification range was between 5.2pmol/L and 400pmol/L.

Major coments :

In order to assess the biological function of PACAP, it is essential to be able to accurately and specifically measure the physiological levels of PACAP in different samples.  The test proposed was the highly specific   The ELISA has been validated to measure PACAP in rodent tissue extracts and in human plasma samples.

Human plasma sera .There is a correlation of the data from the mice used with the amount of human plasma?. Wouldn't another validation be necessary? Describe a little more about -PACAP present in human plasma...

Minor coments : It would be interesting to specify the type of ELISA plate  used in the study in  ELISA sandwich assay as the concentration of monoclonal and policlonal antibodies used  .

What does WT mouse mean? 

For example, table 4

WT1a. What does the (a) stand for?  Add the abbreviations below the table. I believe  it's important to do a general review  review  

Author Response

We are very grateful for the opportunity to revise our manuscript. Hereby, we would like to confirm that all relevant changes have been highlighted in yellow in the new version of the manuscript to visualise them. We would like to thank you for taking the time to review our manuscript

Response to comment 1: 

Response: We wish to direct your attention to the Materials and Methods section of the article, specifically within the Sandwich ELISA protocol subsection. In this section, we already provide information regarding the catalog number of the plates utilized and the concentration of the coating antibody. It's worth noting that we do not specify the concentration of the detection antibody, primarily because it is not purified. For ELISA purposes, we employ this antibody at a concentration of 1:6000, while for immunohistochemistry, a significantly more concentrated titer is employed.

Response to comment 2: 

We would like to point to the materials and methods section of the article, specifically under Animals, here we state that WT means wildtype, hence PACAP proficient mice. We agree that we need to include the abbreviations under each table for clarity and have done so accordingly.

Response to comment 3: 

We concur with the reviewer's perspective, especially regarding the significance of assessing PACAP levels in plasma. Presently, most studies indicate the absence or undetectable levels of PACAP in healthy donors, yet spikes in PACAP have been observed in individuals experiencing migraines. This is precisely where the utility of our assay becomes evident. We have conducted the new ELISA assay on mouse plasma samples and have observed similar recovery rates for spiked samples as those observed in human plasma.

While we acknowledge the potential value in conducting a validation with a broader range of patient samples, we would like to clarify that our current objective is to focus this article solely on the development of the assay itself. In the future, we intend to extend the application of this assay to various patient samples as part of our ongoing research efforts.

Round 2

Reviewer 1 Report

All issues were resolved.